# Light Cigarette Smoking Increases Risk of All-Cause and Cause-Specific Mortality: Findings from the NHIS Cohort Study

**DOI:** 10.3390/ijerph17145122

**Published:** 2020-07-15

**Authors:** Wen Qin, Costan G. Magnussen, Shengxu Li, Lyn M Steffen, Bo Xi, Min Zhao

**Affiliations:** 1Shandong University Hospital, Cheeloo College of Medicine, Shandong University, Jinan 250012, China; qinwen@sdu.edu.cn; 2Menzies Institute for Medical Research, University of Tasmania, Hobart 7000, Australia; costan.magnussen@utas.edu.au; 3Research Centre of Applied and Preventive Cardiovascular Medicine, University of Turku, 20520 Turku, Finland; 4Children’s Minnesota Research Institute, Children’s Hospitals and Clinics, Minneapolis, MN 55404, USA; sli10@tulane.edu; 5Division of Epidemiology and Community Health, School of Public Health, University of Minnesota, Minneapolis, MN 55454, USA; steff025@umn.edu; 6Department of Epidemiology, School of Public Health, Cheeloo College of Medicine, Shandong University, Jinan 250012, China; xibo2007@126.com; 7Department of Nutrition and Food Hygiene, School of Public Health, Cheeloo College of Medicine, Shandong University, Jinan 250012, China

**Keywords:** light smoking, mortality, prospective

## Abstract

Very few studies have examined the association between light cigarette smoking (i.e., ≤5 cigarettes per day) and mortality. The aim of this study was to examine the association of light cigarette smoking with all-cause and cause-specific mortality among adults in the United States. Data were from 13 waves of the National Health Interview Survey (1997 to 2009) that were linked to the National Death Index records through December 31, 2011. A total of 329,035 participants aged ≥18 years in the United States were included. Deaths were from all cause, cancer, cardiovascular disease (CVD) and respiratory disease and were confirmed by death certification. During a median follow-up of 8.2 years, 34,862 participants died, of which 8415 were from cancer, 9031 from CVD, and 2040 from respiratory disease. Compared with never-smokers, participants who smoked 1–2 (hazard ratios (HR) = 1.94, 95%CI = 1.73–2.16) and 3–5 cigarettes (HR = 1.99, 1.83–2.17) per day were at higher risk of all-cause mortality after adjustment for demographic variables, lifestyle factors and physician-diagnosis of chronic disease. The associations were stronger for respiratory disease-specific mortality, followed by cancer-specific mortality and CVD-specific mortality. For example, the HRs (95% CIs) of smoking 1–2 cigarettes per day were 9.75 (6.15–15.46), 2.28 (1.84–2.84) and 1.93 (1.58–2.36), respectively, for these three cause-specific mortalities. This study indicates that light cigarette smoking increases risk of all-cause and cause-specific mortality in US adults.

## 1. Introduction

Cigarette smoking is a global public health issue [1,2] with about 5 million deaths each year attributable to smoking [3]. In the United States (U.S.), the prevalence of cigarette smoking among adults has declined from 20.9% in 2005 to 16.8% in 2014, [4] mainly due to effective tobacco-control policies [5]. However, the risk of deaths from cigarette smoking continues to increase among U.S. adults, especially among men [6].

Evidence from the National Health Interview Survey (NHIS) suggests that among daily cigarette smokers, the percentage who smoked less than 10 cigarettes per day increased from 16% to 27% from 2005 to 2014 [4]. In addition, it is believed that such a low level of smoking exposure might be safe among 10% of U.S. adolescents [7]. Nevertheless, light smoking status is common among smokers who are trying to quit and may be maintained for many years during their lifetime [8,9]. In line, previous studies have suggested light smoking isassociated with increased risk of cancer [10,11], coronary heart disease [12] and stroke. However, limited studies have examined the association between light smoking and risk of mortality [13,14,15]. In addition, participants in most of these studies were either not nationally representative [13,14,15], restricted to male workers [13] or of older age (≥59 years) [14], thus limiting the generalizability of the findings. Importantly, the harmfulness of light smoking has not been given full attention in samples of the general public.

Therefore, we used baseline data from the NHIS collected between 1997 and 2009 linked to data from the National Death Index (NDI) in 2011 to examine the association of cigarette smoking with all-cause and cause-specific mortality among U.S. adults aged ≥18 years.

## 2. Materials and Methods

### 2.1. Study Population

The NHIS is a nationally representative survey of the civilian, noninstitutionalized U.S. population. Detailed information of the NHIS has been described elsewhere [16]. In brief, the NHIS is a cross-sectional survey that has been conducted by the National Center for Health Statistics of the Centers for Disease Control and Prevention since 1957. NHIS uses a stratified, multistage randomized sampling design to collect information using personal household interviews. One sample adult is randomly selected from each household for a detailed interview on health and lifestyle behaviors. NHIS data are de-identified and do not include any protected health information. The data are publicly available and exempt under the ethical board review of the corresponding author’s institution.

As the survey questions of the NHIS underwent major revision in 1997, this study is restricted to NHIS data collected from 1997 to 2009. A total of 366,376 participants aged ≥18 years were included in the preliminary study. Of these, 37,341 were excluded because of missing data on cigarette smoking (*n* = 4301), missing data on covariates (i.e., demographic variables, lifestyle factors and history of physical-diagnosed diseases; *n* = 28,859) or were pregnant (*n* = 4181), leading to a final sample of 329,035 participants.

### 2.2. Mortality Outcomes

All NHIS participants aged ≥18 years from 1997 to 2009 were linked to the NDI records through December 31, 2011, using a probabilistic matching algorithm to determine mortality status [17]. Validation studies have shown that the information on deaths in the NDI records was accurate and in nearly perfect agreement (98.5%) [18,19]. Participants not matched with a death record by December 31, 2011 were considered alive. According to the International Classification of Diseases 10th Revision (ICD-10) codes, study outcomes were defined as: (1) all-cause mortality; (2) CVD-specific mortality (codes I00 to I09, I11, I13, I20 to I51and I60 to I69); (3) heart disease-specific mortality (codes I00 to I09, I11, I13 and I20 to I51); (4) cerebrovascular disease-specific mortality (codes I60 to I69); (5) cancer-specific mortality (codes C00 to C97); and (6) chronic lower respiratory diseases (codes J40 to J47).

### 2.3. Cigarette Smoking

We used NHIS participants’ responses to two questions to define their smoking status (never, former and current smoking): (1) “Have you smoked at least 100 cigarettes in your ENTIRE LIFE?” (yes vs. no) and (2) “Do you NOW smoke cigarettes?” (yes vs. no). According to the responses to these two questions, we categorized participants as never (responded “No” to both questions), former (responded “Yes” to the first question, “No” to the second question) and current smoking (responded “Yes” to the second question). Current smokers were asked “How many cigarettes do you smoke a day?”.Similar with previous publications [13,14,15], we categorized cigarette smoking status into “never”, “former”, “1–2 cigarettes per day”, “3–5 cigarettes per day”, “6–10 cigarettes per day”, “11–20 cigarettes per day”, “21–30 cigarettes per day” and “>30 cigarettes per day”. In this study, participants who consumed≤5 cigarettes per day were considered as light cigarette smokers [7].

### 2.4. Covariates

Demographic variables included sex, age, race or ethnicity (White, Black, Hispanic and others), education level (less than high school, high school and beyond high school), marital status (married; divorced, separated or widowed; and never married). Lifestyle variables included body mass index (BMI, weight divided by height squared, kg/m2), physical activity ((PA) whether or not the participants achieved at least 75 min of vigorous PA or 150 min of moderate PA in one week according to the Physical Activity Guidelines for Americans [20])) and drinking status (lifetime abstainer, former drinker, current light to moderate drinker (1–7 drinks per week for women and 1–14 drinks per week for men) and current heavy drinker (>1 drinks per day for women and >2 drinks per day for men [21,22])). Clinical variables included self-reported physician diagnoses of hypertension, heart disease, stroke, cancer or diabetes.

### 2.5. Statistical Analysis

Differences in baseline participant characteristics between eight categories of cigarette smoking status (never, former smoker, 1–2, 3–5, 6–10, 11–20, 21–30, >30 cigarettes per day) were tested using Chi-square test for categorical variables.

Cox proportional hazards regression model was used to estimate the hazard ratios (HRs) and 95% confidence intervals (CIs) of cigarette smoking status with all-cause and cause-specific mortality, adjusted for potential covariates. The statistical models were sequentially adjusted for covariates to assess for confounding. Model 1 adjusted for demographic factors (sex, age, race or ethnicity, education level, marital status). Model 2 additionally adjusted for lifestyle factors (BMI, PA, alcohol consumption). Model 3 additionally adjusted for clinical variables (history of physician-diagnosed diseases). Both data pooling and meta-analyses were used to calculate summary HRs and 95% CIs.

We also examined the association between cigarette smoking status and all-cause mortality by sex, age group and race/ethnicity. Additionally, in sensitivity analyses, (1) participants with a history of physician-diagnosed diseases were excluded because sick individuals were more likely to quit smoking; (2) individuals who died within the first 2 years (i.e., a 2-year lag) were excluded because the deaths were unlikely caused by smoking within short duration; (3) as the missing data accounted for 9.1% (33,160/366,376) in the total population, multiple imputation for variables with missing values were performed.

To quantitatively evaluate the dose-response association of number of cigarettes per day (as a continuous variable) with all-cause and cause-specific mortality, we performed Cox proportional hazards regression models with penalized splines [23], where the reference value assigned was “never smoking”, using R software v3.3.3 (R Foundation for Statistical Computing, Vienna, Austria), outputting figures with log HR for all-cause and cause-specific mortality on the *y*-axis and current number of cigarettes per day on the *x*-axis.

Survey weights, strata and cluster in the NHIS design were considered in our data analysis. A two-sided *p* < 0.05 was considered statistically significant. All data analyses were performed using SAS version 9.3 (SAS Institute Inc., Cary, North Carolina, USA) and R software v3.3.3 (R Foundation for Statistical Computing, Vienna, Austria).

## 3. Results

The baseline data included 329,035 U.S. adults. Of these, 72,580 were former cigarette smokers (22.1%) and 73,184 were current cigarette smokers (22.2%). Table 1 displays baseline characteristics across the eight categories of cigarette smoking status (never, former smoker, 1–2, 3–5, 6–10, 11–20, 21–30 and >30 cigarettes per day). We observed significant differences in all examined characteristics across the eight categories of cigarette smoking status (all *p* < 0.001; Table 1).

During a median (interquartile range) follow-up of 8.2 (5.0–11.5) years, 34,862 participants died, of which 8415 were cancer-specific deaths, 9031 were cardiovascular disease (CVD)-specific deaths and 2040 were respiratory diseases-specific deaths. There was a dose-dependent association of number of cigarettes per day with all-cause mortality as well as with cause-specific mortality (Table 2). Compared with never smokers, those who currently smoked 1–2 cigarettes per day were at high risk of all-cause mortality (HR = 1.94, 95%CI = 1.73–2.16). The risk was higher among those who smoked 3–5 cigarettes per day (HR = 1.99, 1.83–2.17). Similar trends were observed for cause-specific mortality(cancer mortality: 1–2/day: HR = 2.28 (95%CI = 1.84–2.84), 3–5/day: HR = 2.70 (95%CI = 2.27–3.21); CVD mortality: 1–2/day: HR = 1.93 (95%CI = 1.58–2.36), 3–5/day: HR = 1.96 (95%CI = 1.63–2.35); respiratory disease mortality: 1–2/day: HR = 9.75 (95%CI = 6.15–15.46), 3–5/day: HR = 11.71(95%CI = 8.84–15.52)). (Table 2).

In meta-analysis of the 13 NHIS survey years (Appendix A), results were similar with those from pooled analyses. In sensitivity analyses that excluded participants who died within the first 2 years of follow-up or after exclusion of participants who self-reported physician diagnosis of chronic diseases or that based on imputed data, the results were similar to those shown for the pooled analyses (Appendix A). In the subgroup analyses, the associations tended to be stronger among women, those of older age, and among whites (Appendix A).

Cox models with penalized splines showed that there were curvilinear dose-response associations of number of cigarettes per day (as a continuous variable) with risk of all-cause mortality (Appendix A), cancer-specific mortality (Appendix A), CVD-specific mortality (Appendix A) and respiratory disease-specific mortality (Appendix A), with all *p* < 0.001 for the nonlinear tests. Compared with never smokers, those who reported any number of cigarettes per day had higher risk of all-cause and cause-specific mortality.

## 4. Discussion

To our knowledge, this is the first nationally representative and large-scale study examining the association of light cigarette smoking (i.e., ≤5 cigarettes per day)with risk of all-cause and cause-specific mortality among U.S. adults. Our findings highlight that there is no safe threshold for cigarette smoking, and effective smoking control programs should be conducted to prevent smoking in the general population to reduce risk of mortality.

## 5. Comparison with Previous Studies

Many prospective studies have examined the association of light cigarette smoking with risk of cancer [10,11] and CVD [12]. The NIH-AARP Diet and Health Study of 238,525 healthy adults aged ≥59 years found that compared with never smokers, smokers who smoked <1 and 1–10 cigarettes per day were at higher risk of incident cancer [11]. A recent meta-analysis of 141 cohort studies found that smoking just one cigarette per day increased risk of coronary heart disease and stroke [12]. However, to our knowledge, only few studies have assessed the association of light smoking with risk of mortality [13,14,15] and the findings were inconsistent. In 2002, the Copenhagen City Heart Study of 12,149 adults aged ≥30 years showed that smoking less than 3 g of tobacco per day was not associated with a significantly increased risk of myocardial infarction and all-cause mortality (women: HR = 1.24, 95%CI = 0.79–1.94; men: HR = 1.32, 95%CI = 0.49–3.56). Increased risk seemed to commence from 3 g of tobacco per day [15]. Another prospective study of 25,464 Japanese workers aged 20–61 years showed there was no significant association between a smoking level of 1–10 cigarettes per day and all-cause mortality (HR = 1.51, 95%CI = 0.73–2.94) [13]. The nonsignificant associations in the above two studies could be due to insufficient statistical power in the light smoking groups. Consistent with our findings, the NIH-AARP Diet and Health Study of 290,215 healthy adults aged ≥59 years found occasional smokers (participants who smoked less than 1 cigarette per day) were at higher risk of all-cause mortality (HR = 1.64, 95%CI = 1.07–2.51), as well as several cause-specific mortality outcomes (e.g., lung cancer-specific mortality and CVD-specific mortality) [14]. In addition, we found that the younger participants (18–39 years of age) are more likely to be light cigarette smokers than those in other age categories. This suggests that the younger people are in a more unstable smoking pattern and possibly on a trajectory where an increase in number of cigarettes is more common than the older and more established smokers.

## 6. Public Health Implications

Our findings have several important public health implications. First, our results are consistent with the 2008 Public-Health-Service-sponsored Clinical Practice Guidelines and the 2010 Surgeon General’s Report that no risk-free level of cigarette smoking exists, and even 1–2 cigarettes per day is harmful for health. Thus, it is important that never smokers (especially for young people) should not try to smoke cigarettes, and current cigarette smokers should stop smoking to obtain substantial health benefits. Second, the government should disseminate the dangers of light cigarette smoking via social media. The government should also provide smoking cessation services for current smokers to help them stop smoking. In addition, health professionals and primary care physicians also should inform their patients either not to smoke any dose of cigarettes or to completely stop smoking.

## 7. Study Strengths and Limitations

The main strengths of our study included the very large sample size, relatively longer duration of follow-up, adjustment for potential confounding factors, the use of several sensitivity analyses to test the stability of our findings and the generalizability of our results to the US adults. However, several limitations should be considered. First, information on cigarette smoking was self-reported which might be subject to recall bias. However, validation studies suggest that self-report data are strongly correlated with measured levels of cotinine in blood or urine [24]. Second, cigarette smoking status was only available at baseline, and the change in cigarette smoking status among participants during follow-up might have influenced the findings.

## 8. Conclusions

Our data suggest that light cigarette smoking is associated with higher risk of all-cause and cause-specific mortality compared with never smokers. These data indicate that there is no safe level of cigarette smoking. Our data reinforce the importance of quitting programs aimed at current cigarette smokers across the life-course.

## Figures and Tables

**Table 1 ijerph-17-05122-t001:** Baseline Characteristics according to cigarette smoking status, National Health Interview Survey (NHIS), 1997–2009.

	Cigarette Smoking Status								*p* Value
	Never	Former Smoker	Current Smoker, Cigarettes Per Day						
			1–2	3–5	6–10	11–20	21–30	>30	
*N*	183,269	72,580	6933	10,288	18,266	27,997	5573	4129	
Age, yrs, %									<0.0001
18–39	46.7	20.8	59.6	55.8	51.7	44.2	36.2	26.1	
40–59	33.9	39.8	30.4	33.3	36.8	42.8	50.3	58.8	
≥60	19.5	39.4	10.0	10.9	11.4	13.0	13.4	15.1	
Sex, %									<0.0001
Men	43.8	57.3	56.6	50.4	46.5	55.2	64.6	71.5	
Women	56.2	42.7	43.4	49.6	53.5	44.8	35.4	28.5	
Race/ethnicity, %									<0.0001
White	66.6	82.1	57.6	58.6	69.3	84.3	91.9	92.0	
Black	13.0	7.3	13.6	18.6	16.9	8.4	4.1	4.0	
Hispanic	14.5	7.7	23.5	17.5	9.3	4.6	2.4	2.3	
Other	5.8	2.9	5.2	5.3	4.4	2.6	1.6	1.7	
Education, %									<0.0001
<High school	15.2	16.2	20.0	20.0	21.0	21.1	23.5	30.1	
High school	25.9	29.5	27.8	30.3	35.7	40.5	41.5	39.9	
>High school	58.8	54.3	52.2	49.7	43.3	38.4	35.1	30.0	
Marital status, %									<0.0001
Married	57.6	67.0	45.9	42.0	43.8	48.9	51.7	55.4	
Divorced/separated/widowed	14.9	19.8	14.7	17.7	19.4	21.2	22.6	24.4	
Never married	27.5	13.1	39.4	40.3	36.9	29.9	25.7	20.2	
BMI category, kg/m^2^, %									<0.0001
<18.5	1.9	1.3	2.1	2.9	3.4	2.8	2.2	2.5	
18.5–24.9	39.7	32.2	42.1	43.6	43.8	43.2	39.0	34.8	
25.0–29.9	34.6	39.0	34.2	33.8	31.8	33.4	35.7	35.2	
≥30.0	23.7	27.5	21.6	19.7	21.0	20.7	23.1	27.6	
Physical activity (meeting recommendation), %									<0.0001
No	59.2	58.3	56.1	57.4	64.1	66.6	70.3	75.0	
Yes	40.8	41.7	43.9	42.6	35.9	33.4	29.7	25.0	
Drinking, %									<0.0001
Lifetime abstainer	32.5	8.9	10.3	10.6	10.6	9.4	7.8	9.2	
Former drinker	11.3	23.3	9.0	11.3	14.1	16.3	19.6	22.5	
Light to moderate	53.7	62.3	72.4	68.0	65.9	63.0	55.8	50.2	
Heavy	2.5	5.5	8.3	10.1	9.4	11.3	16.8	18.1	
Physician-diagnosed disease, %									
Hypertension	23.0	36.4	18.4	18.8	20.5	21.8	25.8	31.3	<0.0001
Heart disease	9.1	19.6	8.5	8.1	9.5	10.6	11.8	15.5	<0.0001
Stroke	1.8	4.3	1.6	1.9	2.0	2.3	2.7	3.5	<0.0001
Cancer	5.7	12.3	3.9	4.3	5.2	6.4	7.5	8.0	<0.0001
Diabetes	6.0	11.0	5.1	4.5	5.0	5.0	6.3	9.4	<0.0001

**Table 2 ijerph-17-05122-t002:** Association of cigarette smoking status with risk of all-cause and cause-specific mortality.

	Cigarette Smoking Status, Hazard Ratios (HRs), (95% CIs)							
	Never	Former Smoker	Current Smoker, Cigarettes Per Day					
			1–2	3–5	6–10	11–20	21–30	>30
***N***	183,269	72,580	6933	10,288	18,266	27,997	5573	4129
**All causes**								
No. of deaths	14,646	12,440	513	827	1626	3171	793	846
Model 1	1	1.36 (1.32–1.40)	1.88 (1.68–2.10)	1.96 (1.79–2.13)	2.04 (1.91–2.16)	2.25 (2.14–2.36)	2.70 (2.48–2.94)	3.49 (3.21–3.80)
Model 2	1	1.45 (1.40–1.49)	1.96 (1.76–2.19)	2.04 (1.87–2.22)	2.09 (1.96–2.22)	2.30 (2.19–2.42)	2.77 (2.54–3.02)	3.54 (3.25–3.85)
Model 3	1	1.34 (1.30–1.38)	1.94 (1.73–2.16)	1.99 (1.83–2.17)	2.05 (1.93–2.18)	2.22 (2.11–2.34)	2.63 (2.40–2.88)	3.34 (3.06–3.64)
**Cancer**								
No. of deaths	2898	3159	121	217	452	1020	262	286
Model 1	1	1.73 (1.64–1.84)	2.28 (1.82–2.84)	2.75 (2.32–3.25)	2.59 (2.31–2.91)	3.50 (3.22–3.81)	4.38 (3.79–5.05)	5.68 (4.91–6.58)
Model 2	1	1.80 (1.69–1.91)	2.29 (1.83–2.85)	2.74 (2.31–3.25)	2.54 (2.27–2.86)	3.43 (3.14–3.75)	4.31 (3.73–4.99)	5.54 (4.79–6.42)
Model 3	1	1.67 (1.56–1.77)	2.28 (1.84–2.84)	2.70 (2.27–3.21)	2.49 (2.22–2.80)	3.29 (3.01–3.60)	4.02 (3.47–4.66)	5.26 (4.54–6.10)
**CVD**								
No. of deaths	4120	3183	121	176	387	683	186	175
Model 1	1	1.23 (1.16–1.30)	1.88 (1.53–2.30)	1.93 (1.60–2.31)	2.33 (2.02–2.68)	2.39 (2.16–2.64)	3.16 (2.65–3.75)	3.59 (3.01–4.29)
Model 2	1	1.34 (1.27–1.43)	2.01 (1.64–2.46)	2.05 (1.70–2.46)	2.47 (2.14–2.85)	2.52 (2.27–2.78)	3.35 (2.81–3.99)	3.73 (3.11–4.48)
Model 3	1	1.21 (1.14–1.28)	1.93 (1.58–2.36)	1.96 (1.63–2.35)	2.40 (2.07–2.77)	2.36 (2.11–2.64)	3.09 (2.56–3.72)	3.46 (2.89–4.15)
**Heart disease**								
No. of deaths	3053	2543	91	136	307	573	159	154
Model 1	1	1.29 (1.21–1.37)	1.76 (1.38–2.24)	1.86 (1.53–2.26)	2.47 (2.10–2.91)	2.53 (2.26–2.83)	3.53 (2.91–4.27)	3.94 (3.26–4.75)
Model 2	1	1.40 (1.32–1.50)	1.89 (1.48–2.40)	1.99 (1.63–2.42)	2.65 (2.24–3.14)	2.69 (2.40–3.02)	3.76 (3.10–4.56)	4.11 (3.39–4.98)
Model 3	1	1.25 (1.17–1.33)	1.83 (1.43–2.33)	1.90 (1.56–2.31)	2.56 (2.16–3.04)	2.51 (2.22–2.84)	3.47 (2.83–4.25)	3.78 (3.12–4.58)
**Cerebrovascular disease**								
No. of deaths	1067	640	30	40	80	110	27	21
Model 1	1	1.07 (0.95–1.20)	2.43 (1.57–3.77)	2.26 (1.51–3.39)	2.02 (1.57–2.59)	1.94 (1.56–2.42)	1.92 (1.26–2.94)	2.39 (1.48–3.88)
Model 2	1	1.19 (1.05–1.35)	2.57 (1.66–3.99)	2.32 (1.55–3.49)	2.04 (1.59–2.63)	1.98 (1.59–2.46)	1.99 (1.30–3.04)	2.44 (1.50–3.97)
Model 3	1	1.10 (0.97–1.24)	2.49 (1.64–3.80)	2.23 (1.49–3.33)	1.99 (1.54–2.57)	1.83 (1.46–2.30)	1.75 (1.13–2.69)	2.35 (1.44–3.81)
**Respiratory diseases**								
No. of deaths	294	1020	32	77	163	298	70	86
Model 1	1	6.66 (5.66–7.84)	9.91 (6.17–15.93)	12.66(9.56–16.77)	15.32(12.22–19.22)	16.57(13.53–20.29)	21.10(15.33–29.04)	35.11(26.10–47.23)
Model 2	1	7.55 (6.39–8.93)	9.97 (6.26–15.89)	12.17(9.19–16.12)	13.92(11.09–17.48)	15.26(12.49–18.64)	20.05(14.56–27.61)	32.38(24.14–43.42)
Model 3	1	6.96 (5.88–8.23)	9.75 (6.15–15.46)	11.71(8.84–15.52)	13.79(10.96–17.35)	14.53(11.86–17.79)	18.52(13.44–25.52)	30.52(22.63–41.15)

Model 1: Adjusted for sex, age, and race/ethnicity, education and marital status. Model 2: Model 1 plus additionally adjusted for body mass index, alcohol intake and physical activity. Model 3: Model 2 plus additionally adjusted for physician-diagnosed diseases (hypertension, heart disease, stroke, cancer and diabetes).

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
