# Peer review of "Light Cigarette Smoking Increases Risk of All-Cause and Cause-Specific Mortality: Findings from the NHIS Cohort Study"

_ijerph, 2020, doi:10.3390/ijerph17145122_

Round 1

Reviewer 1 Report

This is a retrospective analysis of the National Health Interview Survey (NHIS). The study confirm well-known results (e.g. WHO almost always underline that there is no safe level of tobacco use).

The authors should focus on general concussions ("there is no safe level of smoking") rather, than specific data (X - light smokers dies X-years earlier).

There are some other comments that should be addressed

Methods: "Table 1 displays baseline characteristics 134 across the 8 categories of cigarette smoking status (never, former smoker, 1-2, 3-5, 6-10, 11-20, 21-30, and >30 cigarettes per day)." Please justify why do the authors prepare 6 different categories (1-2 up to > 30). Do they use the currently available scale? any scientific justification? or subjective categorization?

Results: Findings from table 2 should be described including more details

Discussion: The current version of the discussion is too short, and should be more comprehensive. Please underline the novelty of this study and potential practical implications.  

Conclusions: Current versions repeat well-known observations. Please provide to re-write this section based of the findings from this study

Author Response

This is a retrospective analysis of the National Health Interview Survey (NHIS). The study confirms well-known results (e.g. WHO almost always underline that there is no safe level of tobacco use).

1. The authors should focus on general conclusions ("there is no safe level of smoking") rather, than specific data (X - light smokers dies X-years earlier).

Response: We thank for the reviewer’s suggestion. We have focused on general conclusion in the Abstract and Discussion section.

There are some other comments that should be addressed

2. Methods: "Table 1 displays baseline characteristics across the 8 categories of cigarette smoking status (never, former smoker, 1-2, 3-5, 6-10, 11-20, 21-30, and >30 cigarettes per day)." Please justify why do the authors prepare 6 different categories (1-2 up to > 30). Do they use the currently available scale? any scientific justification? or subjective categorization?

Response: The categories of cigarette smoking status are based on previous publications on this topic. We have cited these publications for scientific justification.

3. Results: Findings from table 2 should be described including more details

 Response: We have provided more descriptions about table 2.

4. Discussion: The current version of the discussion is too short, and should be more comprehensive. Please underline the novelty of this study and potential practical implications.  

Response: We have provided one paragraph on public health implications of our findings.

5. Conclusions: Current versions repeat well-known observations. Please provide to re-write this section based of the findings from this study

Response: We have re-written the conclusion section based on our findings.

Reviewer 2 Report

  1. Please define the “light smoking”.
  2. Line 47: “In addition, it is believed that such a low level of smoking exposure might be safe.[7]” Your conclusion doesn't seem to correspond with the quoted literature.
  3. Line 121: ”3) as the missing data accounted 121 for 9.1% (33,160/366,376) in the total population, multiple imputation for variables with missing 122 values were performed and we obtained similar results (data not shown).” If possible, please to attach the results.
  4. Line 146: “In meta-analysis of the 13 NHIS survey years (Table S1)”. I didn’t see the table S1.

Author Response

1. Please define the “light smoking”.

Response: We have defined “light smoking” in the Methods section.

2. Line 47: “In addition, it is believed that such a low level of smoking exposure might be safe.[7]” Your conclusion doesn't seem to correspond with the quoted literature.

Response: Thanks. We have re-written this sentence clearly.

3. Line 121: ”3) as the missing data accounted 121 for 9.1% (33,160/366,376) in the total population, multiple imputation for variables with missing values were performed and we obtained similar results (data not shown).” If possible, please to attach the results.

Response: We have provided results based on imputed data in Table S4.

4. Line 146: “In meta-analysis of the 13 NHIS survey years (Table S1)”. I didn’t see the table S1.

Response: We did present the results of meta-analysis in table S1 in our previous version. We did not know why the reviewer did not see this table. You may contact the editor for this table.

Reviewer 3 Report

This is a neat little paper that is also well written.

The objective studied has been visited by other researchers before but the strengths with this study is the representativity and the many categories of number of cigaretts per day that gives a very nice dose response relationship.

COMMENT.

In table 1 it looks as if the 1-2 cig/day are younger that the other categories. It could be that the younger subjects are in a more unstable smoking pattern and possibly on a trajectory where an increase in number of cigarettes is more common than with the older and more established smokers. Please address that  in the discussion or  limitations sections.

Author Response

This is a neat little paper that is also well written.

1. The objective studied has been visited by other researchers before but the strength with this study is the representatives and the many categories of number of cigarettes per day that gives a very nice dose response relationship.

Response: We thank for the reviewer’s positive comments.

COMMENT.

2. In table 1 it looks as if the 1-2 cig/day are younger that the other categories. It could be that the younger subjects are in a more unstable smoking pattern and possibly on a trajectory where an increase in number of cigarettes is more common than with the older and more established smokers. Please address that in the discussion or limitations sections.

Response: We agree. We have added this point in the Discussion section.

Round 2

Reviewer 1 Report

Please provide more informative conclusion, which is based on own findings, and focus on the study group. "This study indicates that there is no safe level of cigarette smoking." - this is a well-known fact

Author Response

Please provide more informative conclusion, which is based on own findings, and focus on the study group. "This study indicates that there is no safe level of cigarette smoking." - this is a well-known fact.

Response: Thank you. We have changed conclusion to "This study indicates that light cigarette smoking increases risk of all-cause and cause-specific mortality in US adults."